# Tyrosine Kinase Inhibitor Post-Allogeneic Stem Cell Transplantation in Adult Philadelphia-Positive B-Acute Lymphoblastic Leukemia: State of the Art and Future Directions

**DOI:** 10.3390/cimb47020129

**Published:** 2025-02-18

**Authors:** Martina Canichella, Paolo de Fabritiis

**Affiliations:** 1Hematology, St. Eugenio Hospital, ASL Roma2, 00144 Rome, Italy; paolo.defabritiis@aslroma2.it; 2Department of Biomedicina e Prevenzione, Tor Vergata University, 00133 Rome, Italy

**Keywords:** Philadelphia-positive B-cell acute lymphoblastic leukemia (Ph+ ALL), allogeneic stem cell transplantation (HSCT), tyrosine kinase inhibitors (TKI), dasatinib, ponatinib, blinatumomab, measurable residual disease (MRD), *IKZF1^plus^*

## Abstract

In a scenario characterized by continuous improvement in outcomes, Philadelphia chromosome-positive (Ph+) ALL, once considered a biologically defined subtype with one of the poorest prognoses, now includes patients achieving long-term survival even without allogeneic stem cell transplantation. First-line therapy is increasingly adopting a chemo-free approach, combining tyrosine kinase inhibitors (TKIs) with immunotherapy—specifically blinatumomab—which has resulted in high rates of complete molecular responses and improved survival outcomes. Within this paradigm shift, the allocation to transplantation is becoming increasingly selective and genomically oriented, focusing on patients with particularly unfavorable prognostic and predictive factors. For patients undergoing transplantation, maintenance therapy with TKIs has emerged as one of the most important strategies to reduce the risk of relapse. However, there remains considerable uncertainty regarding which patients benefit most from this approach, the optimal TKI agents, dosing strategies, and the duration of maintenance therapy. In this review, we aim to consolidate the available evidence on this topic, analyzing it in the context of the most recent clinical experiences.

## 1. Introduction

Philadelphia chromosome-positive (Ph+) ALL derives from the reciprocal translocation between chromosome 9 and chromosome 22, leading to the chimeric *BCR:ABL1* fusion gene.

In Ph+ ALL, the *BCR::ABL1* fusion gene encodes for the 190-Kd protein (p190) in 65% of cases, while the 210-Kd protein (p210) occurs in 35% of patients [1]. Because p210 is more common in chronic myeloid leukemia (CML), it is difficult to distinguish at diagnosis whether acute leukemia derives from a blastic crisis of CML or from Ph+ ALL de novo. p190 or p210 are responsible for the hyperactivation of the tyrosine kinase cascade, which leads to the leukemogenesis process, supporting the proliferation and survival of blast cells.

Ph+ ALL represents the most common B-ALL subtype in adults, comprising 20–25% of all genetic subgroups. The incidence rises with advancing age; while uncommon in pediatric populations, it is observed in more than 50% of patients aged over 50 years [2]. Historically, Ph+ ALL is associated with very poor prognosis: before the introduction of imatinib in 2000, the long-term survival rate for these patients was around 10–20% [3,4,5,6]. Imatinib, a first-generation TKI, marked the beginning of a new era in the treatment of Ph+ ALL. It led to the introduction of a strategy based on TKIs that was initially combined with more or less intensive chemotherapy regimens, with or without allogeneic hematopoietic stem cell transplantation (HSCT); more recently, it has involved chemotherapy-free protocols (TKIs combined with blinatumomab) [6,7,8,9,10,11,12,13,14,15,16,17]. This paradigm shift significantly enhanced treatment responses and improved long-term outcomes. Efforts to improve the prognoses of these patients have also been driven by a deeper understanding of pathogenesis and underlying molecular lesions, many of which are associated with specific prognostic impact, as well as advancements in measurable residual disease (MRD) monitoring [18,19]. In particular, MRD can be monitored using various methodologies, including flow cytometry and molecular techniques such as PCR (both real-time and digital droplet PCR), and, more recently, next-generation sequencing (NGS) [20]. Nowadays, chemotherapy-free regimens are emerging as the efficacy strategy, inducing deep and early molecular responses. These advances have led to a critical reassessment of patient eligibility for HSCT, a decision that has paradoxically become increasingly complex. Among all post-transplant maintenance strategies in Ph+ ALL, the use of TKI has been well-established [21]. Nonetheless, issues remained unsolved for those patients, for whom both the setting (prophylactic, if MRD is negative; preemptive, when MRD became positive without frank hematological relapse) and the generations of TKI need to be better defined to ameliorate maintenance therapy post-HSCT. The current European and American guidelines recommend the use of TKI post-HSCT, but they preceded the widespread use of dasatinib and ponatinib in combination with blinatumomab [22,23]. Furthermore, there is a lack of randomized prospective clinical trials that definitively address this topic. In the setting of post-HSCT, some aspects should be taken into consideration: the effects of the graft-versus-leukemia (GVL) balance; the risk of graft-versus-host disease (GvHD); the optimal TKI dosage that should be optimized to minimize side effects without compromising therapeutic efficacy. The aim of this review is to provide an update on the state of the art regarding TKI maintenance post-HSCT, highlighting the latest evidence and addressing the unresolved question on the feasibility of discontinuing TKI therapy in maintenance post-HSCT.

### Imatinib

Imatinib mesylate, with or without chemotherapy and generally followed by HSCT, has revolutionized outcomes for Ph+ ALL patients. Before the introduction of imatinib in 2000, the survival of Ph+ ALL patients was dismal, with an OS rate of about 10–20% [3,4,5,6].

Imatinib functions by binding to the *BCR::ABL1* kinase domain, located in an inactive conformation in a pocket reserved for the ATP binding site. Imatinib inhibits the BCR-ABL protein by binding to the ATP-binding pocket through two mechanisms: either via a lock-and-key mechanism or through the formation of hydrogen bonds. Imatinib specifically binds to an inactive conformation of BCR-ABL, characterized by a closed state of the enzyme, which is incapable of binding ATP. Imatinib is active on several proteins such as Abl, c-kit, PDGFR, and c-fms kinases [24].

Initially, imatinib was administered in combination with chemotherapy, leading to improvements in 5-year disease free survival (DFS) and in overall survival (OS) of around 50% [9,25,26,27,28,29,30,31]. In this context, the GIMEMA group introduced a pioneering new strategy based on a chemo-free induction phase with TKI, steroids, and central nervous system (CNS) prophylaxis. As will be discussed here, this strategy will prove to be successful and will serve as the basis for the development of the TKI–blinatumomab immunotherapy, which is currently in use. This chemo-free approach achieved a rate of complete remission (CR) very close to 100% and a low toxicity with no early deaths during induction. In the context of this era, HSCT represented the best consolidation therapy for eligible patients and was the treatment that achieved a higher OS, mostly in CR1.

Retrospective studies have shown that patients who received imatinib after HSCT had a better prognosis than those who did not receive it. In these experiences, imatinib was administered either as prophylaxis or preemptive at doses ranging from 300 to 600 mg. The study that considered the largest number of patients was reported by Brissot et al. on behalf of the EBMT Leukemia Working Party [25]. Of 473 patients with Ph+ ALL, 157 received TKI after allo-HSCT; 124 patients in this group received imatinib prophylactically. The multivariate analysis showed that TKI post-HSCT resulted in a lower incidence of relapse (HR = 0.40; *p* = 0.01) and improved OS and leukemia-free survival. Table 1 summarizes the characteristics and results of these retrospective series and prospective studies.

Regarding prospective study, the most important one, because it is a prospective randomized clinical study, was conducted by Pfiefer et al. [26]. In this study, imatinib was administered either as a prophylactic (n = 26) or in a preemptive strategy (n = 29) following allo-HSCT, with treatment planned for a duration of 1 year. The results revealed that prophylactic imatinib use was associated with a reduced incidence of molecular relapse compared to its use at the initial detection of BCR-ABL transcripts (40% versus 69%; *p* = 0.046). The long-term outcomes for OS and EFS at 5 years were 80.1% versus 74.5% (*p* = 0.84) and 72.1% versus 53.7% (*p* = 0.89), respectively. However, these findings did not achieve statistical significance.

Wassmann et al. enrolled 27 patients with Ph+ ALL who underwent either allogeneic (n = 24) or autologous (n = 3) SCT; the study demonstrated that preemptive (MRD-positive) administration of imatinib (400 mg with possible escalation to 600–800 mg daily in patients with persistent MRD-positive) is associated with conversion to MRD negativity in more than 50% of cases [27]. Furthermore, MRD negativity achieved within three months of imatinib therapy was statistically associated with improved DFS and OS compared to patients who remained persistently MRD-positive. Notably, among the 10 patients who achieved MRD negativity and subsequently discontinued imatinib, 3 experienced molecular relapses, with 2 of these patients subsequently reverting to MRD-negativity.

Carpenter et al., for the first time, reported the results of a prospective trial in which imatinib was administered prophylactically from the time of engraftment until 1 year after HSCT at a dose of 200 mg/day or more [28]. Twenty-two enrolled patients (fifteen treated with Ph+ ALL and seven treated with CML) received imatinib after HSCT, with primary end-point during the first 90 days of treatment after HSCT being the safety. The most frequently reported adverse events were grade 1–3 nausea, emesis, and elevated serum transaminase levels (two adult patients discontinued treatment before 90 days due to transaminase elevations). Interestingly, at the time of the drafting, 17 patients are alive in complete molecular remission and 15 have been followed-up with after imatinib discontinuation at 1 year (median of 0.45 years -range, 0–1.6 years). Only two patients are scheduled to discontinue imatinib within 3 months. The authors concluded that imatinib can be administered safely in the early post-myeloablative allogeneic HCT setting, with a dose intensity comparable to that employed in frontline therapeutic protocols.

Ram and colleagues reported the results of non-myeloablative conditioning HSCT in 51 patients treated with Ph+ ALL in CR [29]. Eighteen received imatinib (at a dose of 400–600 mg/daily) post-engraftment. The 3-year OS rate was 62%; for the subgroup without evidence of MRD at the time of transplantation, the OS rate was 73%.

Ottman et al. demonstrated that the outcome of patients undergoing HSCT in CR or >CR1 and receiving imatinib either as prophylaxis or preemptive is superior among those who undergo HSCT in CR [30].

Ribera et al. reported a prospective phase II trial in which imatinib, at a dose of 400 mg/day, was administered post-transplantation in 12/27 patients for a median duration of 3.9 months, regardless of MRD status [9]. Among these, 10 patients discontinued the drug, with only 2 cases being attributed to drug-related side effects. With a median follow-up of 4.1 years, the OS and DFS for allograft patients was only 30%. This study revealed that the duration of post-transplant imatinib administration was limited, probably related to transplantation complications rather than drug-specific toxicity.

Indeed, Chen et al. reported statistically superior outcomes for 62 patients who received imatinib post-HSCT independently of disease status compared with those who did not (n = 20) (5-year OS and EFS were 86.7% versus 34.3% and 81.5% versus 33.5%, respectively, *p* < 0.001) [31]. Notably, 8 of 14 patients (57%) converted from MRD positivity to negativity during imatinib therapy.

Finally, the randomized study conducted by Pfeifer et al. contributed to the field by demonstrating that imatinib administered as a prophylactic (n = 26) resulted in a lower incidence of the recurrence of transcript compared with the preemptive group (n = 29) (40% versus 69%; *p* = 0.046) [26]. The 5-year OS and EFS for the two groups resulted in 80.1% versus 74.5% and 72.1% versus 53.7% but did not achieve statistical significance.

Although comparative studies between imatinib maintenance and newer-generation TKIs are not currently available, the clinical experience with imatinib highlighted that the duration of maintenance therapy should not be less than one year or at least one year following the achievement of MRD negativity; it has also been demonstrated that the efficacy of imatinib post-HSCT has better results for those patients allografted in CR1.

Taken together, these results suggest a benefit of imatinib maintenance post-HSCT with a favorable safety profile. This favorable tolerability could be particularly advantageous in cases where newer TKIs induce toxicity or become intolerant in the post-HSCT context.
cimb-47-00129-t001_Table 1Table 1Prospective and retrospective studies with imatinib after HSCT in Ph+ ALL.Author, YearsType of StudyPatients(n)StrategyStart(mo)Dose(mg/d)MedianDuration(Range)OSDFSAEPfeifer, 2013 [26]Prospective54Prophylactic26 >CR1 = 31.6400–6001 year5-year OS 80%DFS 5 years—83.9%CR = 81% at 30.3 mRelapse = 2%, >CR1 = 40%GI intolerance n = 6Hematotoxicity n = 5GVHD n = 2Preemptively29 >CR1 = 22.3400–6001 year5-year OS74.5%DFS 5 years—60.4%CR = 78% at 32.4 mrelapse 17%>CR1 = 40%Chen, 2012 [31]Prospective82Preemptive and prophylactic2.3300–400BCR-ABL-negative-3 mo5-yearOS = 86.7%5-year DFS = 81.5%5-year relapse = 10.2%>CR1 OS HR = 2.7Neutropenia = 43%Thrombocytopenia = 43% Edema = 23%Nausea = 35%Ram, 2011 [29]Prospective51Prophylactic1/2200–60012 mo (3–50 m)3-yearOS = 62%;For MRD = 73%Decreased mortalityHR = 0.3 *p* = 0.03, relapse = 21%,>CR1 = 1.5 yr OS = 25%Grade II GVHD 53%;GI-2, Pleural effusion—1Ribera, 2010 [9]Prospective13Prophylactic3.9200–4001 yearscheduled; 9months was average for receiving treatment 1.7-yearOS = 30%1.5-year DFS = 30%Relapse (n = 3) Severe chronic GVHD(n = 2) Grade 3–4 toxicity (n = 2, hematologic in 1 patient and GI in the other)Ottmann, 2009 [30]Prospective40Prophylactic and preemptive1.5–3400–600-1.5 yearsOS = 92%EFS 1.5 years—92%Relapse—0%GI toxicity (n = 5)GVHD (n = 3)Carpenter, 2007 [28]Prospective27Prophylactic1260–400121.3 yearsat OS = 80%Relapse rate 13%Emesis, N, increased AST/ALT level, anemia, edema, diarrheaGrade 3/4—17% Tx terminated—16%ThrombocytopeniaNeutropeniaAnorexiaWassmann, 2005 [27]Prospective27Preemptively4.4400–60012 mo from negative PCR1–2-year OS for MRD ve (n = 14) = 80–100%1–2-year DFS for MRD = 54.5–91%1-year DFS and OS for MRD-positive(n = 13) = 8% and 23%GI discomfort n = 1Edema and weight gain n = 1Brissot, 2015 [25]Retrospective157Prophylactic2.7--5-year OS: HR = 0.42(0.23–0.76)*p* = 0.0045-year LFS: HR = 0.44 (0.26–0.74)*p* = 0.0025 years RIHR = 0.4 *p* = 0.01-OS: overall survival; DFS: disease-free survival; CR: complete remission; MRD: measurable residual disease; GI: gastrointestinal.

## 2. Second-Generation TKIs

### 2.1. Nilotinib

Nilotinib is a second-generation TKI that is not currently approved for the treatment of Ph+ ALL. In vitro, nilotinib is 30 times more potent than imatinib and is active against many imatinib-resistant BCR-ABL mutations [32]. Other tyrosine kinases inhibited by nilotinib are c-Kit and PDGFR. Available studies that have investigated its role also included patients with chronic myeloid leukemia (CML). Nilotinib has been administered in doses ranging from 200 to 300 mg twice daily. Despite the small number of patients with Ph+ ALL, these studies showed that nilotinib as maintenance post-HSCT demonstrated safety, with a lower number of side effects, and has been found to be associated with improved outcomes [33,34,35].

### 2.2. Dasatinib

Since 2006, when dasatinib was administered in the front-line setting, available data on the long-term outcomes of Ph+ ALL patients have further increased. Unlike imatinib, dasatinib is able to inhibit SFK-dependent r and is 325 times more potent than imatinib [36]. It binds to both the active and inactive conformations of the ABL kinase domain and also binds PDGFR and c-KIT.

In particular, the MDACC tested dasatinib with hyper-CVAD regimen and reported a CR rate of 96% and a 5-year OS rate of 46% [37]. Notably, the GIMEMA study group, following the strategy of a chemo-free approach based on the combination of dasatinib and blinatumomab, tested in phase II, LAL2116, D-ALBA [38]. Blinatumomab is a bi-specific antibody belonging to the family of BiTE antibodies. BiTE antibodies link T cells and tumor cells, triggering the signaling cascade of the T cell receptor complex by binding to the CD3 receptor. In particular, the target antigen of blinatumomab is CD19, which is expressed on the B-cell lineage [39]. In the D-ALBA trial, blinatumomab and dasatinib were administered to 63 patients; at 18 months, the OS and DFS were 95% and 88%, respectively. With a longer follow-up of 53 months, these results were confirmed with OS, DFS, and EFS rates of 80.7%, 75.8%, and 74.6%, respectively. The D-ALBA trial helped point out that the rate of complete molecular remission (CMR) increased with the numbers of blinatumomab cycles (from 29% after one cycle to 60% after two cycles) [40]. Interestingly, 50% of patients received allografts, showing a very low rate of transplant mortality; this is probably explainable by the absence of previous chemotherapy toxicity [38]. Indeed, the molecular studies of D-ALBA revealed that the signatures associated with the worst outcome were represented by *IKZF1^plus^* cases in which the deletion of *IKZF1* co-exists with the deletion of *CDKN2A-B* and *PAX5*, underling a genomic instability of these cases [41]. *IKZF1* and *PAX5* are genes that play a critical role in the development of B-ALL; in particular, *IKZF1* is involved in B-cell lymphoid differentiation, while *CDKN2A-2B* is essential in the proliferation of leukemic cells [42]. The discovery of this poor molecular profile led to its identification at diagnosis in the subsequent GIMEMA LAL2820 trial, as mentioned below. The experiences with dasatinib at doses ranging from 50 to 100 mg post-HSCT are very few, even though the results are encouraging: Caocci et al. reported 10 patients who received imatinib or dasatinib in the front-line and subsequently received dasatinib as maintenance post-HSCT (median duration, 15 months) [43]. Two of three patients who were MRD-positive after transplant successfully converted to molecular negativity with dasatinib. In another study by Czyz et al., 19 patients received dasatinib (duration of 11 months), regardless of their MRD status [44]. After a median follow-up of 3 years, the OS and leukemia-free survival (LFS) rates were 87% and 88%, respectively. Fourteen of fifteen patients (93%) who were MRD-positive after transplant successfully converted to MRD negativity. Subsequently, six out of nine patients reported by Maher et al. [45] received dasatinib in post-HSCT, showing a 100% LFS at 1.4 years. Table 2 reported the most important studies with second-generation TKIs in post-HSCT.

### 2.3. Ponatinib

Ponatinib, the third-generation TKI, was introduced in 2010 in the treatment of Ph+ ALL. Ponatinib is a potent pan-*BCR::ABL1* inhibitor which is able to induce deeper molecular remission and overcome the TKI resistance due to the onset of T315I mutation (gatekeeper mutation) [46,47]. The T315I mutation of the ABL1 kinase domain is responsible for resistance to imatinib and second-generation TKIs, leading to treatment failure. The T315I mutation is involved replacing threonine with isoleucine; this results in the loss of a hydrogen bond that is necessary for the effective and high affinity of first- and second-generation TKIs. Additionally, isoleucine introduces steric hindrance, obstructing the proper positioning of TKIs within the ATP-binding site but still allowing access to ATP. The effectiveness of ponatinib depends on its structure, which is similar to that of ATP. The pleiotropic effect on several tyrosine kinase pathways is responsible for the off-target effect of ponatinib, which mostly impacts the cardiovascular system. Nowadays, these adverse events are more manageable, as mentioned below [48,49].

The incidence of T315I mutation increased up to 75% in patients (mostly elderly patients), who relapsed after treatment with first- and second-generation TKIs [50,51]. The MDACC investigated the role of ponatinib in combination with hyper-CVAD, showing high CMR and an OS rate of 75% at 6 years with a median follow-up of 80 months [52,53]. Notably, this study evidenced a trend of better OS rates in patients who were not submitted to HSCT upon their first remission. Indeed, the same study contributed to managing the cardiotoxicity of ponatinib, reducing the dose based on the response from 45 mg to 30 mg in CR and 15 mg in CMR, and underlined the importance of 12 intrathecal (IT) for CNS prophylaxis.

Both meta-analysis and comparative studies have demonstrated the superiority of ponatinib regimens compared with the previous TKIs, emphasizing the concept that patients who manage to achieve complete molecular remission (CMR) at 3 months show a better OS compared to those who remained positive [52]. Indeed, a randomized prospective study compared ponatinib (30 mg/day) with imatinib (600 mg/day) plus mild chemotherapy in adult patients with newly diagnosed Ph+ ALL. The ponatinib arm demonstrated a higher rate of MRD negativity. In terms of outcomes, the median EFS was not reached in the ponatinib group [54]. The toxicity profiles of the two TKIs were found to be comparable. Based on the success of dasatinib, ponatinib was also studied at the onset of the disease; the MDACC group conducted a study on 44 patients treated with ponatinib and blinatumomab for a total of five cycles. The CMR rates recorded after one and five courses were 64% and 85%, respectively. Indeed, patients who achieved MRD negativity (n = 25) with NGS (sensitivity of 10^−6^) achieved 95% rates for both PFS and OS after 3 years. Notably, only one patient received HSCT in CR1 due to a persistently detectable *BCR::ABL1* transcript [53]. The GIMEMA study group is also investigating the role of ponatinib in combination with blinatumomab in the phase III trial 2820, which is open to recruitment [55]. The preliminary results were presented at the 66th ASH meeting, showing promising results with a well-tolerated profile: at a median follow-up of 6.4 months (range 0.1–32.3), the estimated 18-month OS is 91.6%.

In the post-HSCT setting, there are few experiences with ponatinib compared with the other TKIs.

Chen et al. reported the outcomes of 26 Ph+ ALL patients who harbored T315I mutation and were treated with ponatinib after allograft (12 as prophylactic and 7 as preemptive) [56]. The 5-year DFS, OS, and cumulative incidence of relapse (CIR) in patients receiving prophylactic ponatinib were 81.5%, 91.7%, and 18.5%. Indeed, this study demonstrated the feasibility and the manageable toxicity of ponatinib.

A multicenter observational retrospective study by Candoni et al. reported results for 46 patients with Ph+ ALL who underwent allo-HSCT and received ponatinib prophylactically (n = 26) or preemptively (n = 22) [57]. All these patients were allografted in CHR and 46% of those were in CMR. The relapse-free survival (RFS) of the entire cohort was 71% at 60 months; RFS was significantly better in the group who received ponatinib in prophylaxis (95%) compared to preemptively (57%). This study demonstrated that ponatinib also presented a good safety profile, particularly for those cases treated in prophylaxis (Table 3).

## 3. Discussion

The advent of TKIs revolutionized the treatment of Ph+ ALL. Since the introduction of imatinib, research has identified more innovative compounds to overcome resistance due to the onset of mutation, mostly on the kinase domain of ABL. Among TKIs, ponatinib resulted in impressive clinical results due to its activity for T315I mutation.

With regard to the use of TKI in the post-HSCT for Ph+ ALL, we underlined some unsolved issues. Several retrospective and observational studies showed that the administration of TKIs post-HSCT is commonly administered in clinical practice. The GITMO study group reported an observational study in which 40% of patients (178 of 441) received TKIs, with 53% (n = 94) of them being preemptive [58]. In the MDACC study group, a total of 97 patients received TKIs (71 as prophylaxis and 26 as preemptive). The results showed that the TKI maintenance correlated with a lower rate of hematological relapse and improved PFS [59]. Subsequently, EBMT and ASCTC published guidelines which recommend the administration of TKIs after allograft; in particular, EBMT exposed nine points that nowadays present some limitations [25]. The experts recommended the use of TKI to all allograft patients, both as prophylaxis and preemptive when driven by MRD monitoring. The experts recommended imatinib at 600 mg/daily and the shift to second-generation TKI can be driven by imatinib toxicity or resistance. Regarding the duration of treatment, expert recommendations suggest at least 12 months of MRD negativity if the transplant was performed in CR1 and indefinite continuation if the transplant was performed in >CR2. However, this aspect remains unclear and is not standardized across different centers. Nonetheless, clinical practice tends to favor continuing treatment for at least 2 years in cases where allogeneic transplantation was performed with MRD negativity, and treatment continues indefinitely for high-risk patients. These recommendations appear to be unsuitable with the current progress of Ph+ ALL. As mentioned above, the breaking points in Ph+ ALL are represented by the introduction of newer generations of TKIs, their combination with blinatumomab, the discovery of novel molecular lesions with prognostic implications, and finally a more precise monitoring technique for MRD, such as DD-PCR or NGS [60,61]. These advances can allow us to improve the outcomes of these treatments and to refine our choices for post-induction treatment (HSCT). Indeed, with the advent of ponatinib and blinatumomab regimen, the early molecular response (detected with deeper technology) has emerged as one of the most important factors of survival. This paradigm shift has led to the choice of HSCT being more difficult. The allocation to HSCT, therefore, will be reserved only for cases with poor prognostic factors such as the presence at diagnosis of *IKZF1^plus^* or the persistent positivity of MRD at 3 months. These high-risk patients will be treated with maintenance and the choice of TKI will fall on newer TKI generations. The second important issue regards the possibility of stopping TKI maintenance. EBMT suggested that TKI treatment should be sustained for 12 months under continuous MRD negativity, and indefinitely for patients undergoing HSCT in ≥CR2. Few experiences have been reported for TKI discontinuation in allografted patients. Nakasone et al. reported a study in which 92 allo-HCT patients received TKI for >3 months after HSCT [62]. A total of 39 patients received TKI as prophylaxis, while 53 received the treatment on the basis of MRD positivity, with no differences between the two groups. TKIs were stopped in 48 patients and discontinuation, expressed as a time-dependent covariate, was not associated with subsequent hematological relapse. In the TKI-stop group, the TKI administration for >6 months correlated with superior RFS, leading the researchers to draw the conclusion that TKI can be stopped safely for those with persistent MRD negativity. Unfortunately, observational studies showing the management of different hematological centers in deciding whether or not to continue long-term TKI administration are still lacking, as are prospective randomized studies that will be important in defining the characteristics of patients who could potentially be candidates for stopping TKIs. In this setting, the GIMEMA LAL2217 (NCT03318770) study is an observational clinical study that recently closed enrollment; they will reveal a post-frontline (dasatinib plus blinatumomab) therapeutic strategy. Indeed, in the GIMEMA LAL2116 study, the choice of treatment after the dasatinib and blinatumomab cycles is based on clinical decision. These data will reveal whether a subgroup of patients, either undergoing or not undergoing allogeneic transplantation, has discontinued treatment with TKIs.

## 4. Conclusions

Currently, the lack of prospective clinical trials undoubtedly limits the evidence for a definitive benefit of using TKIs in post-transplant maintenance. However, experiences from various study groups suggest a potential advantage, and in clinical practice, TKIs are administered to post-HSCT patients whenever feasible. The introduction of chemotherapy-free regimens in the frontline setting has, paradoxically, made the decision to proceed with HSCT more complex, while decreasing the percentage of patients who will undergo allo-HSCT. This choice now requires careful consideration of additional genetic lesions and early treatment responses. In the near future, it is likely that only very high risk patients, such as those with genomic abnormalities or persistent MRD positivity, will undergo HSCT in CR1. These patients will be the primary candidates for maintenance therapy with second- or third-generation TKIs. A critical issue to address will be determining the optimal duration of TKI maintenance. Advances in molecular monitoring techniques for MRD will play a pivotal role in identifying patients who may safely discontinue TKI therapy, while enabling close monitoring for disease recurrence. These approaches will further refine and personalize post-HSCT management strategies.

Advances in molecular monitoring techniques for MRD will play a pivotal role, helping to identify patients who may safely discontinue TKI therapy and enabling close monitoring for disease recurrence. These approaches will further refine and personalize post-HSCT management strategies.

## Figures and Tables

**Table 2 cimb-47-00129-t002:** Second-generation TKI (dasatinib and nilotinib) post-HSCT in Ph+ ALL.

Author, Years	Type of Study	Patients (n)	Type of TKI	Strategy	Start (mo)	Dose (mg/d)	Median Duration (Range)	OS	DFS	AE
Caocci, 2012 [43]	Retrospective	10	DAS	Maintenance	3.78 (1.63–9)	50–100	15 mo (3–75)	Median OS: 22 mo (8–87)		Diarrhea (grade 1) Pleural effusion (grade 2) Hematologic toxicity (grade 2)
Czyz, 2016 [44]	Retrospective	15	DAS	Preemptive	7 (1–22)	100	11 mo (2–39)	3-year OS: 87%	3-year LFS: 88% Relapse rate: 10%	Heme toxicities n = 2 Pleural effusion n = 1, Infection n = 2 Liver toxicities n = 1
		2	DAS	Prophylactic	5 (2–6)	100				
Varda-Bloom, 2017 [33]	Prospective	16	NILO	Prophylactic	1.8 (1.1–6.67)	200–300	> or =3 mo	OS: 10.2–97.1 mo (median 35.5 mo)	DFS: 10.2– 97.1 mo (median 34.1 mo)	Sepsis = 1 Nocardiosis = 1 Brain bleeding = 1
Shimoni, 2015 [34]	Prospective	22	NILO	Prophylactic	1.26 (1–5.26)	200–300	12 mo (0.3–65)	2-year OS: 69%, 4-year: 46% Compared with entire transplant pt with 2-year OS: 55%, 4-year: 36%	2-year PFS: 56% 4-year PFS: 42% Compared with entire transplant pt with 2-year PFS: 45%; 4-year PFS: 34%	Hematotoxicity = 2 Abdominal pain and vomiting = 4 QTc prolongation = 1 CVA = 1
Nagler, 2009 [35]	Prospective	11	NILO 1.26	Prophylactic	(1–1.63)	200 mg	7.5 mo (4–12)	OS-77% at 7.5 months	Relapse: 0%	

OS: overall survival; DFS: disease-free survival; PFS: progression-free survival.

**Table 3 cimb-47-00129-t003:** Ponatinib post-HSCT in Ph+ ALL.

Author, Year	Type of Study	Patients (n)	Strategy	Start (mo)	Dose (mg/d)	Median Duration (mo)	OS	DFS	AE
Candoni, 2024 [57]	Retrospective	26	Prophylactic	7.4 months (range 2–63)	30 mg/day (range 15–45)	22 months (range 2–100)	-	5-year RFS 95%	-
		22	Preemptive					5-year RFS 57%	
Chen H, 2022 [56]	Retrospective	12	Prophylactic	-	-	-	5 years OS 91.7%	5-year DFS 81.5%	69.4% of patients mild AE
		7	Preemptive				5-year OS 46%	5-year DFS 39.8%	

OS: overall survival; DFS: disease-free survival.

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
