# Peer review of "Tyrosine Kinase Inhibitor Post-Allogeneic Stem Cell Transplantation in Adult Philadelphia-Positive B-Acute Lymphoblastic Leukemia: State of the Art and Future Directions"

_cimb, 2025, doi:10.3390/cimb47020129_

Round 1
Reviewer 1 Report
Comments and Suggestions for Authors
In the submitted manuscript, Canichella and colleague provide a comprehensive review of tyrosine kinase inhibitors (TKIs) used as post-allogeneic stem cell transplantation (HSCT) maintenance therapy in Ph+ acute lymphoblastic leukemia (ALL). Overall, I think the authors provides a thorough and concise review of the existing literature in the field. The manuscript is well-structured and written. I believe the scope of the paper will be of interest to a broad readership of the journal.
However, several points need to be improved before final publication:
1. Could the authors provide some discussion of potential biomarkers for predicting response to TKI maintenance? Perhaps including a section on emerging biomarkers and their potential role in personalizing maintenance therapy would be helpful.
2. Could the authors provide a more in-depth discussion of how the emergence of chemotherapy-free induction regimens (e.g TKI + blinatumomab) might change the current treatment strategies?
3. Along the line of point 3, could the authors discuss the optimal duration of TKI maintenance post-HSCT, and how should this be determined?
4. The tables presented in the manuscript need better formatting. With the current scheme, they appear rather unpolished and hastily prepared.
Author Response
Referee 1
In the submitted manuscript, Canichella and colleague provide a comprehensive review of tyrosine kinase inhibitors (TKIs) used as post-allogeneic stem cell transplantation (HSCT) maintenance therapy in Ph+ acute lymphoblastic leukemia (ALL). Overall, I think the authors provides a thorough and concise review of the existing literature in the field. The manuscript is well-structured and written. I believe the scope of the paper will be of interest to a broad readership of the journal.
However, several points need to be improved before final publication:
Comment 1: Could the authors provide some discussion of potential biomarkers for predicting response to TKI maintenance? Perhaps including a section on emerging biomarkers and their potential role in personalizing maintenance therapy would be helpful.
Response 1: Thank you for your insightful comment. At present, there are no biomarkers available for clinical practice that can predict the response to TKI maintenance therapy. However, it is true that monitoring response to TKI treatment, both pre- and post-transplant, is based on the assessment of minimal residual disease (MRD), which can be performed using flow cytometry or molecular biology techniques such as PCR or NGS. In the post-transplant setting, disease progression can also be monitored through chimerism analysis, which can be assessed using FISH or STR methodologies.
We chose not to include a dedicated section on these aspects, as they represent standard clinical practice and fall outside the specific scope of this review. Nonetheless, we appreciate your suggestion and recognize the value of further exploration into emerging biomarkers for personalizing maintenance therapy in future research.
Comment 2: Could the authors provide a more in-depth discussion of how the emergence of chemotherapy-free induction regimens (e.g TKI + blinatumomab) might change the current treatment strategies?
Response 2: Thank you for your suggestion. The topic of chemotherapy-free induction regimens in Ph+ B-ALL is particularly close to my heart, both because I trained at Sapienza University of Rome, where I had the opportunity to work firsthand on this type of strategy alongside Prof. Sabina Chiaretti with the GIMEMA protocols, and because it was a central focus of my PhD thesis.
Initially, Prof. de Fabritiis and I considered including a section on this topic. However, as the primary aim of this review is to focus on post-HSCT maintenance, we felt that delving into chemotherapy-free induction regimens would take us too far beyond the scope of the manuscript.
We truly appreciate your comment and acknowledge the importance of this emerging approach, which undoubtedly warrants detailed discussion in dedicated studies or reviews.
Comment 3. Along the line of point 3, could the authors discuss the optimal duration of TKI maintenance post-HSCT, and how should this be determined?
Response 3: Thank you for your observation. We have added a clarification regarding the duration of maintenance therapy in the text.
Comment 4: The tables presented in the manuscript need better formatting. With the current scheme, they appear rather unpolished and hastily prepared.
Response 4: Thank you for your observation. You are absolutely right that the table may appear somewhat minimalistic; however, its design is intentional. The aim was to provide a concise, impactful summary that facilitates easy reading and quick reference to the clinical studies and the results they have yielded.
We believe this format enhances clarity and accessibility, but we are open to making adjustments should further refinements be deemed necessary.
.
Reviewer 2 Report
Comments and Suggestions for Authors
Martina Canichella and Paolo de Fabritiis performed a literature review and wrote this manuscript on the evidence of using TKIs in the post-HSCT setting in Ph+ALL. This is an important topic and although the literature review is thorough and well done; the lack of prospective randomized trials precludes for any definitive or strong evidence-based recommendations. Major revisions would be necessary to improve the manuscript:
Major points:
1. I would change the title and emphasize in the manuscript that this review focuses on adult Ph+ALL since there are differences in the management of pediatric vs. adult Ph+ALL. Specifically, the use of TKI post-HSCT is less routinely used in pediatric Ph+ALL than adult Ph+ALL.
2. I would also emphasize the lack of prospective randomized trials that are statistically well-powered to define the role and benefit of TKI use in the post-HSCT setting in adult Ph+ALL. Consequently, I would highlight the above point upfront and soften the recommendations/conclusions regarding the clear benefit of TKI use post-HSCT. (For example, the conclusion -line 324- undoubtedly improve outcome should be modified.
3. In the section on Imatinib use post-HSCT, reorganization of data presentation would enhance the clarity of the manuscript. I have the following recommendations:
a. I would move up the presentation of the trial by Pfeifer et al., and provide more details about this trial since this is the only prospective randomized trial on this topic. Subsequently, instead of listing consecutive retrospective studies, I would present the ones to support the authors' conclusions on imatinib use post-HSCT that 1) imatinib maintenance duration should be not less than one year or 2) at least one year following the achievement of MRD negativity, and 3) its efficacy is greater in patients allografted in CR1.
4. To improve clarity to the readers, the authors should define early in the manuscript prophylactic vs. pre-emptive TKI use. In addition, whenever applicable, they should precise the MRD assay that was used and threshold to define MRD positivity vs. negativity.
5. In the Ponatinib section, the authors should include and briefly discussed the results of the randomized phase 3 PhALLCON trial that compares ponatinib vs. imatinib in frontline adult Ph+ALL. Jabbour EKantarjian HMAldoss I, et al. Ponatinib vs Imatinib in Frontline Philadelphia Chromosome–Positive Acute Lymphoblastic Leukemia: A Randomized Clinical Trial. JAMA. 2024;331(21):1814–1823. doi:10.1001/jama.2024.4783
6. In the Discussion and future questions, the authors should highlight the different unknown and unanswered questions or challenges in the treatment of Ph+ALL in the future:
a. The percentage of CR1 HSCT will likely decrease in the era of immunotherapy + newer generation TKIs in the frontline setting.
b. It is unknown whether patients who received upfront immunotherapy and newer generation of TKI will benefit from TKI post-HSCT in CR1 ?
c. Discuss different strategies on how disease can be monitored in the setting of TKI post-HSCT.
Minor points:
1. Please correct the following typos:
a. Line 145: Pfeifer et al. instead of Pfizer at al.
b. Line 247: 66th instead of 66 degrees
c. Line 325: frontline
d. Line 329: CR1 HSCT
Comments on the Quality of English LanguageThere are no major orthographic deficiencies in English, but would benefit from a English language review to improve the flow and clarity of sentences.
Author Response
Referee 2
Martina Canichella and Paolo de Fabritiis performed a literature review and wrote this manuscript on the evidence of using TKIs in the post-HSCT setting in Ph+ALL. This is an important topic and although the literature review is thorough and well done; the lack of prospective randomized trials precludes for any definitive or strong evidence-based recommendations. Major revisions would be necessary to improve the manuscript:
Major points:
Comment 1 I would change the title and emphasize in the manuscript that this review focuses on adult Ph+ALL since there are differences in the management of pediatric vs. adult Ph+ALL. Specifically, the use of TKI post-HSCT is less routinely used in pediatric Ph+ALL than adult Ph+ALL.
Response 1 Thank you for your valuable observation. We fully agree with your comment and have revised the title to specify that the focus is on the post-transplant setting in adult patients with Ph+ B-ALL. We believe this change provides greater clarity and aligns more precisely with the scope of the manuscript.
Comment 2 I would also emphasize the lack of prospective randomized trials that are statistically well-powered to define the role and benefit of TKI use in the post-HSCT setting in adult Ph+ALL. Consequently, I would highlight the above point upfront and soften the recommendations/conclusions regarding the clear benefit of TKI use post-HSCT. (For example, the conclusion -line 324- undoubtedly improve outcome should be modified.
Response 2 We agree, and thank you for the suggestion. We have revised the main text accordingly.
Comment 3. In the section on Imatinib use post-HSCT, reorganization of data presentation would enhance the clarity of the manuscript. I have the following recommendations:
- I would move up the presentation of the trial by Pfeifer et al., and provide more details about this trial since this is the only prospective randomized trial on this topic. Subsequently, instead of listing consecutive retrospective studies, I would present the ones to support the authors' conclusions on imatinib use post-HSCT that 1) imatinib maintenance duration should be not less than one year or 2) at least one year following the achievement of MRD negativity, and 3) its efficacy is greater in patients allografted in CR1.
Response 3: We agree, and thank you for the suggestion. We have made changes to the main text following your recommendations.
Comment 4. To improve clarity to the readers, the authors should define early in the manuscript prophylactic vs. pre-emptive TKI use. In addition, whenever applicable, they should precise the MRD assay that was used and threshold to define MRD positivity vs. negativity.
Response 4: Thank you for this suggestion. We have clarified the difference between prophylactic and preemptive maintenance strategies in the text
Commnet 5. In the Ponatinib section, the authors should include and briefly discussed the results of the randomized phase 3 PhALLCON trial that compares ponatinib vs. imatinib in frontline adult Ph+ALL. Jabbour E, Kantarjian HM, Aldoss I, et al. Ponatinib vs Imatinib in Frontline Philadelphia Chromosome–Positive Acute Lymphoblastic Leukemia: A Randomized Clinical Trial. JAMA. 2024;331(21):1814–1823. doi:10.1001/jama.2024.4783
Response 5: Thank you for the suggestion. We have included this important study in the text and updated the bibliography accordingly.
Comment 6. In the Discussion and future questions, the authors should highlight the different unknown and unanswered questions or challenges in the treatment of Ph+ALL in the future:
- The percentage of CR1 HSCT will likely decrease in the era of immunotherapy + newer generation TKIs in the frontline setting.
- It is unknown whether patients who received upfront immunotherapy and newer generation of TKI will benefit from TKI post-HSCT in CR1 ?
- Discuss different strategies on how disease can be monitored in the setting of TKI post-HSCT.
Response 6: Thank you for the suggestions. We have revised the text and updated the bibliography
Minor points:
- Please correct the following typos:
- Line 145: Pfeifer et al. instead of Pfizer at al.
- Line 247: 66th instead of 66 degrees
- Line 325: frontline
- Line 329: CR1 HSCT
Reviewer 3 Report
Comments and Suggestions for Authors
Ph+ Allograft-negative (Ph+) ALL patients are achieving long-term survival without allogeneic stem cell transplantation. First-line therapy combines tyrosine kinase inhibitors (TKIs) with immunotherapy, resulting in high rates of complete molecular responses and improved survival. Maintenance therapy with TKIs is crucial for patients undergoing transplantation to reduce relapse risk. However, there is uncertainty regarding optimal TKI agents, dosing strategies, and maintenance therapy duration.
1. The title was "Tyrosine Kinase Inhibitors Post-Allogeneic Stem Cell Transplantation: State-of-Art and Future Directions"; however, the authors appear to overlook the strategies involving TKIs with or without post-HCT. As an oncologist, I am interested in understanding the role of TKIs during the post-HCT phase.
2. The description of the three generations of TKIs should be streamlined, as it is not the focus of this manuscript.
3. The authors could create a table to compare the two groups that utilize TKIs following allogeneic HCT versus those that do not. This would enhance the clarity of the review for the audience regarding the role of TKIs after allogeneic HCT in cases of Ph+ ALL.
Author Response
Referee 3
Ph+ Allograft-negative (Ph+) ALL patients are achieving long-term survival without allogeneic stem cell transplantation. First-line therapy combines tyrosine kinase inhibitors (TKIs) with immunotherapy, resulting in high rates of complete molecular responses and improved survival. Maintenance therapy with TKIs is crucial for patients undergoing transplantation to reduce relapse risk. However, there is uncertainty regarding optimal TKI agents, dosing strategies, and maintenance therapy duration.
Comment 1: The title was "Tyrosine Kinase Inhibitors Post-Allogeneic Stem Cell Transplantation: State-of-Art and Future Directions"; however, the authors appear to overlook the strategies involving TKIs with or without post-HCT. As an oncologist, I am interested in understanding the role of TKIs during the post-HCT phase.
Response 1: Thank you for your interest
Comment 2. The description of the three generations of TKIs should be streamlined, as it is not the focus of this manuscript.
Response 2: Thank you for this observation. Initially, we did not consider including this aspect, but upon the suggestion of the CIMB editor, we decided to provide more depth on molecular aspects in order to meet the journal's requirements.
Comment 3. The authors could create a table to compare the two groups that utilize TKIs following allogeneic HCT versus those that do not. This would enhance the clarity of the review for the audience regarding the role of TKIs after allogeneic HCT in cases of Ph+ ALL.
Response 3: Thank you for this observation. Initially, we did not consider including this aspect, but upon the suggestion of the CIMB editor, we decided to provide more depth on molecular aspects in order to meet the journal's requirements.
Round 2
Reviewer 3 Report
Comments and Suggestions for Authors
The authors failed to address my inquiries. They are merely attempting to fulfill the CIMB editor's suggestion to enhance the depth on molecular aspects to align with the journal's requirements. I believe I did not play a part in the new comments for this paper.
Comments on the Quality of English Language
The reviewer declared no conflicts of interest.